# MSFT: Mitigating Spurious Correlations in Text Classification via Feature Induction in Embedding Layers and Tensor Stretching

## Abstract

Text classification stands as one of the core tasks in natural language processing (NLP). However, it has long been plagued by the issue of spurious correlations, where models tend to learn associations between non-informative or spurious input features and class labels, thereby constraining generalization capability and classification performance. To tackle this challenge, we formalize the notion of a 'semantic centroid' and, leveraging this construct, propose a novel plugin termed MSFT (**M**itigating **S**purious Correlations in Text Classification via **F**eature Induction in Embedding Layers and **T**ensor Stretching). The MSFT plugin first computes a semantic centroid—encapsulating the global semantic information of the entire dataset—by aggregating all embedding tensors within the dataset. During model training, it alleviates spurious correlations through tensor distance stretching in the embedding space, specifically targeting the subset of data that drives the formation of such spurious associations. Designed for seamless integration with existing classification architectures, MSFT boasts strong generality and scalability. We conduct experiments on two representative categories of language models: BERT-style masked language models (MLMs) and autoregressive large language models (LLMs; e.g., GPT-2). Extensive experiments across multiple datasets demonstrate that our plugin effectively mitigates the detrimental impacts of spurious correlations while consistently improving classification performance. Notably, it achieves or even surpasses state-of-the-art (SOTA) benchmarks in spurious correlation mitigation for text classification, regardless of the underlying model architecture.

## 1 Introduction

Text classification has long stood as a foundational research topic in natural language processing (NLP), with widespread applications spanning diverse subtasks including sentiment analysis and topic categorization. Since the rise of deep learning, a succession of influential text classification models has been developed, encompassing early architectures such as TextCNN Chen (2015), later BERT-based frameworks Devlin et al. (2019), and the state-of-the-art CAPO model Zehle et al. (2025).

Spurious correlations persist as a key challenge in text classification, stemming primarily from models inadvertently learning non-informative feature-label associations. When such training-induced correlations fail to generalize to test data, model performance often degrades substantially Ye et al. (2024), as illustrated in Figure 1.

In recent years, a growing body of researchers has focused on addressing the challenge of spurious correlations in text classification tasks, developing a range of methodologies: **Data augmentation** modifies inputs to learning algorithms to enhance the generalization capability and diversity of the training data distribution; **Learning strategy development** centers on designing in-learning mechanisms that mitigate over-reliance on spurious correlations while improving model generalization to novel data; **Representation learning** seeks to strengthen the model's comprehension of latent variable relationships while enhancing robustness against spurious correlations Ye et al. (2024).

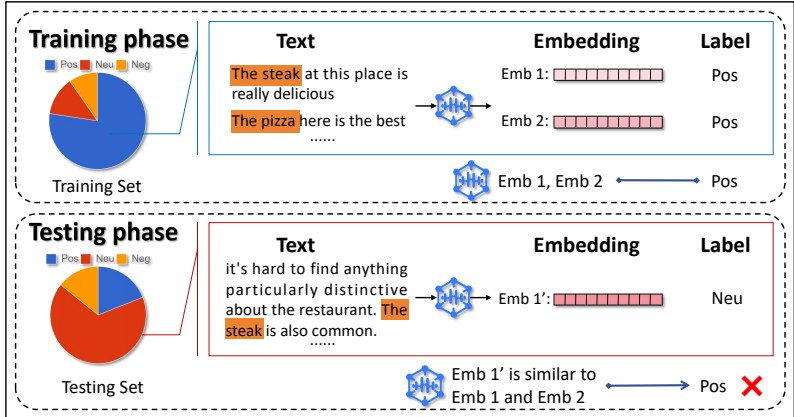

Figure 1: **How Spurious Correlations Mislead Model Learning.** Consider a cluster comprising embeddings such as $\mathbf{Emb}_1$ and $\mathbf{Emb}_2$, where the majority of training instances are labeled positive. The model thus learns to associate embeddings semantically similar to $\mathbf{Emb}_1/\mathbf{Emb}_2$ with the positive class. During testing, when encountering a new embedding $\mathbf{Emb}'_1$ with high semantic similarity to $\mathbf{Emb}_1$ and $\mathbf{Emb}_2$, the model erroneously predicts a positive label—contrary to its true neutral (neu) label. **This error arises because embedding tensors of clustered texts exhibit extreme semantic similarity, creating misleading feature associations Stein et al. (2024)**. Our proposed solution mitigates such errors by disentangling these features through tensor stretching.

In recent years, research on spurious correlations has gradually increased, and attention has been paid to spurious correlations in various downstream tasks of natural language understanding, with some breakthroughs being made. Focusing on the field of text classification, data augmentation methods attempt to enhance model performance by adding new data to datasets Kaushik et al. (2019); Zeng et al. (2020); Fern & Pope (2021); Wang & Culotta (2021); Wen et al. (2022); Bhattacharjee et al. (2023), but they incur high costs in data generation and labeling Wang & Culotta (2021), while providing limited improvement in mitigating spurious correlations Ye et al. (2024). In terms of learning strategies, various loss function-based approaches Sagawa et al. (2019); Chew et al. (2023); Zhang et al. (2023) and phased learning strategy Liu et al. (2021); Kirichenko et al. (2022); LaBonte et al. (2023) have been proposed; however, experiments indicate that no single loss function can comprehensively and effectively address all types of spurious correlations, and their practicality remains to be improved Zhou et al. (2024). The phased learning strategy also leads to an increase in training duration. Meanwhile, research in representation learning on spurious correlations is still in its early stages, and feature disentanglement is considered a crucial means to address spurious correlations Zhou et al. (2023); Principe et al. (2024). A more comprehensive introduction is provided in the Appendix A.3.

The design of the embedding layer is crucial and exerts a significant influence on the learning of spurious correlations Bugliarello et al. (2021). Bias in the embedding of tokens can lead to the acquisition of spurious associations from the very beginning of the learning process Bolukbasi et al. (2016). To the best of our knowledge, there has been no in-depth research to date addressing the mitigation of spurious correlations at this layer.

Meanwhile, downsampling guided by human priors Li et al. (2024) —an important approach for mitigating spurious correlations Zhou et al. (2023) —cannot be effectively incorporated into current loss functions or phased training methods due to its relatively superficial processing nature.

A simple, practical, and plug-and-play solution capable of effectively addressing various types of spurious correlations remains to be proposed. To bridge this research gap and integrate embedding layer optimization with downsampling principles, we propose MSFT[1], a universal plugin that operates via three key steps: (1) computing a global semantic centroid; (2) partitioning excess samples; (3) applying standard training while transforming outlier samples through centroid stretching to

---

[1]Code will be released publicly

mitigate spurious patterns in novel embedding dimensions. Our contributions are summarized as follows:

- We present the MSFT plugin—the first plug-and-play solution for mitigating spurious correlations at the embedding layer in text classification.

- We conduct evaluations across six pre-trained models and three representative, diverse datasets. The plugin enables most models to outperform their original baselines, with several even surpassing state-of-the-art (SOTA) performance. These results underscore its effectiveness and generalizability.

## 2 RELATED WORK

Most research addressing spurious correlations in text classification has focused on enhancing model performance by introducing novel loss functions. For instance, the DRO Sagawa et al. (2019) method improves generalization on worst-group performance by implementing distributionally robust optimization in over-parameterized neural networks, while emphasizing the critical role of regularization in this process. NFL Chew et al. (2023) approach analyzes local neighborhoods to understand and mitigate spurious correlations in text classification, proposing a family of NFL loss functions to counteract the impact of various types of spurious associations on model learning. Similarly, the A2R Zhang et al. (2023) framework defines and optimizes a causal probabilistic loss function to select necessary and sufficient explanatory texts, thereby enhancing both model interpretability and generalization capability. In terms of training strategies, DFR Kirichenko et al. (2022) effectively reduces the model's reliance on spurious correlations and significantly enhances robustness by retraining only the last layer of the neural network. Similarly, LLR LaBonte et al. (2023) introduces a method called Selective Last-Layer Fine-Tuning (SELF), which requires no group annotations and only minimal class annotations, yet substantially improves model robustness on worst-group performance. However, Shortcut Maze Zhou et al. (2024) empirically demonstrates that existing methods for mitigating spurious correlations remain insufficient to comprehensively mitigate the impacts of various types of spurious associations. These approaches are also accompanied by issues such as prolonged training duration.

The embedding layer, as the initial processing stage in model training, plays a critical role in shaping the learning of spurious correlations Bolukbasi et al. (2016); Bugliarello et al. (2021). Effectively addressing biases at this layer may offer a novel perspective for mitigating spurious associations.

The downsampling methodology incorporating human priors Li et al. (2024) has been demonstrated as an effective strategy for mitigating spurious correlations Zhou et al. (2023). However, it has not been adequately integrated into existing mitigation frameworks and is generally employed merely as a baseline approach, which represents a significant underutilization.

Anchored in these insights, our MSFT plugin achieves unsupervised feature disentanglement informed by human priors directly within the embedding layer. It operates exclusively on raw text-label pairs, requiring neither data augmentation nor supplemental annotations. By holistically leveraging all available samples and globally restructuring their embeddings, our method provides comprehensive mitigation of spurious correlations.

## 3 METHOD

Spurious correlations emerge when a specific feature maintains an overly strong association with data labels, a phenomenon rooted in imbalanced label distributions within the corresponding cluster Zhou et al. (2023). Based on this observation, we characterize the over-represented samples in the dataset as high-risk instances—these samples are particularly susceptible to being captured by the model through shortcut learning mechanisms. That is, within a cluster, all instances whose quantity under their respective labels exceeds that of the minimally represented label are defined as 'excess samples'. To address this issue, our MSFT plugin proactively applies regularization to these redundant samples, thereby mitigating the model's tendency to internalize spurious correlation features and effectively achieving downsampling at the embedding layer.

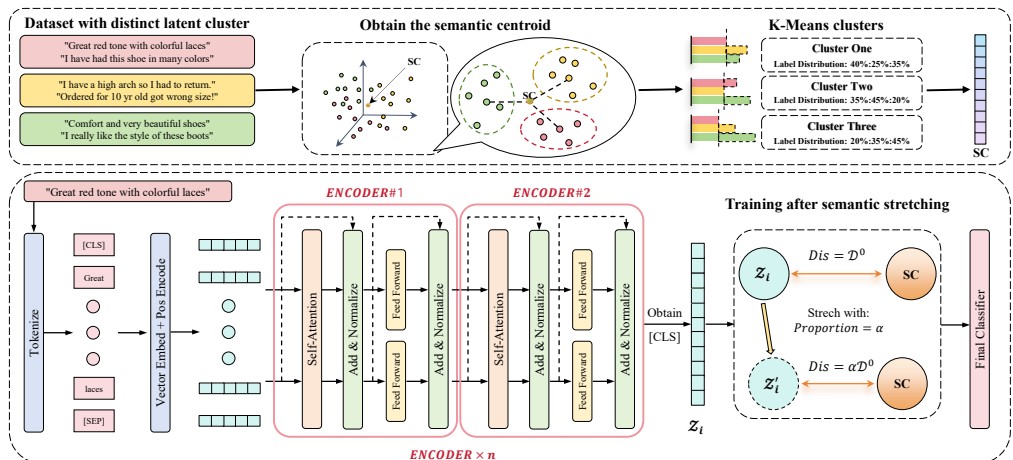

Figure 2: **Plugin Implementation Flowchart.**1)Identify the 'semantic centroid'(SC) for each dataset via clustering on original data. 2)For each cluster in the raw dataset: Identify the label with the smallest quantity. Align other labels to this quantity, identifying excess samples and record them. 3)For each excess samples point, transform it into a word embedding tensor, adjust its distance to the identified 'semantic centroid' in semantic space by a set SR, and then feed it to the model for further training.

First, we preprocess the original dataset. For each dataset, we employ the K-Means algorithm to compute its semantic centroid and identify excess samples. Next, we extract the embedding tensor output by the model's embedding layer and apply semantic stretching transformations to it. For excess samples, we individually stretch their embedding tensors toward the 'semantic centroid' within the semantic space before feeding them into the model for further training, as illustrated in Figure 2. A summary of the mathematical notations used in this chapter is provided in Appendix Table 6.

## 3.1 ACQUISITION OF SEMANTIC CENTROID

For a given dataset $\mathcal{D}$, we begin by encoding the dataset $\mathcal{D}$ using the model's encoder layer to derive the set of encoded representations $\mathcal{Z} = \{f_\theta(x) \mid x \in \mathcal{D}\}$, where $f_\theta$ denotes the encoding function. Next, we determine the optimal number of clusters $m^*$ by applying the K-Means algorithm with the elbow method Thorndike (1953)—a technique used to identify the optimal cluster count in unsupervised K-Means learning:

$$m^* = \text{ElbowMethod}(\text{K-Means}(\mathcal{Z})), \tag{1}$$

then obtain the corresponding cluster centers $\mathcal{C} = \{c_1, \ldots, c_{m^*}\}$. The semantic centroid $\mathbf{s}$ is subsequently computed as the mean tensor of these cluster centers(Please note that the process of computing the semantic centroid is performed only once during the initial training phase and does not undergo further iterations):

$$\mathbf{s} = \frac{1}{|\mathcal{C}|} \sum_{c \in \mathcal{C}} c. \tag{2}$$

Meanwhile, we obtain the clustering results of the embedding layer, denoted as *cluster groups* $\{C_1, ..., C_{m^*}\}$.

## 3.2 IDENTIFY EXCESS SAMPLES

Within each cluster group $C_i$, the label distribution demonstrates substantial imbalance. For example, within a single cluster, positive labels may dominate ($|\mathcal{L}_{C_i}^+| \gg |\mathcal{L}_{C_i}^-|$) while negative labels

are scarce, whereby the spurious correlation within $C_i$ is manifested. We formally define the set of *excess samples* as:

$$\mathcal{E}_{C_i} = \{x \mid x \in C_i, \text{count}(y_x) > \min_{y \in Y} \text{count}(y)\},$$

which corresponds to instances exceeding the minority class size within each cluster. These excess samples $\mathcal{E}_{C_i}$—high-risk instances prone to inducing shortcut learning in the model—are explicitly annotated to construct a newly labeled dataset $\mathcal{D}'$ that highlights these problematic subsets. Our goal is to mitigate spurious correlations through targeted processing of this specifically annotated subset.

### 3.3 CENTRIPETAL STRETCHING IN EMBEDDING SPACE

Upon obtaining the semantic centroid $\mathbf{s}$, during the fine-tuning phase of the pre-trained model, we intercept the embedding tensor $z_i$ generated for each sample $x_i$. Using the annotation labels, we determine whether $x_i$ belongs to the excess sample set $\mathcal{E}$. For samples identified as excess, their embedding tensors are stretched along the direction of the semantic centroid according to a predefined stretching ratio:

$$z_i' = \begin{cases} \mathbf{s} + \alpha(z_i - \mathbf{s}) & \text{if } x_i \in \mathcal{E}, \\ z_i & \text{otherwise}, \end{cases} \tag{3}$$

where $\alpha$ denotes the stretching ratio (a hyperparameter), and $\mathcal{E} = \bigcup_{i=1}^{k} \mathcal{E}_{C_i}$ represents the union of all excess samples across clusters. The modified embeddings $z_i'$ are then fed into subsequent layers of the pre-trained model for continued training. The complete algorithmic workflow is detailed in Algorithm 1.

---

**Algorithm 1** Semantic Stretching for Fine-Tuning

---

**Input:** $\mathcal{M}_{\text{pre}}$: pre-trained model $\mathcal{D} = \{(x_i, y_i)\}_{i=1}^{N}$: dataset $\alpha$: stretching ratio $m^*$: number of clusters (optional)

**Output:** $\mathcal{M}_{\text{finetuned}}$: fine-tuned model

**Phase 1: Semantic Analysis**

1. Extract embeddings: $\mathcal{Z} \leftarrow \{f_{\mathcal{M}_{\text{pre}}}(x_i) \mid x_i \in \mathcal{D}\}$

2. Cluster centers and Cluster groups: $\{c_1, \ldots, c_{m^*}\}, \{C_1, \ldots, C_{m^*}\} \leftarrow \text{K-Means}(\mathcal{Z}, m^*)$

3. Compute 'semantic centroid' through Cluster centers: $\mathbf{s} \leftarrow \frac{1}{m^*} \sum_{j=1}^{m^*} \text{centroid}(c_j)$

4. Identify excess samples: **for** *each cluster $C_j$* **do**

    $n_{\min}^j \leftarrow \min_{y \in Y} |\{x \in C_j \mid y_x = y\}|$ $\mathcal{E}_{C_j} \leftarrow \emptyset$ **for** *each label $y \in Y$* **do**

        $\mathcal{E}_{C_j} \leftarrow \mathcal{E}_{C_j} \cup \{\text{random subset of } \{x \in C_j \mid y_x = y\} \text{ with size } \max(0, |\{x \in C_j \mid y_x = y\}| - n_{\min}^j)\}$

    $\mathcal{E} \leftarrow \bigcup_{j=1}^{k} \mathcal{E}_{C_j}$

**Phase 2: Training with Stretching** **while** *not converged* **do**

    **for** *each batch $\mathcal{B} \subset \mathcal{D}$* **do**

        **for** *each $x_i \in \mathcal{B}$* **do**

            $z_i \leftarrow f_{\mathcal{M}_{\text{pre}}}(x_i)$ **if** $x_i \in \mathcal{E}$ **then**

                $z_i' \leftarrow z_i + \alpha(z_i - \mathbf{s})$ ;        // Stretch toward semantic centroid

            **else**

                $z_i' \leftarrow z_i$ ;                // Keep original embedding

            Compute loss $\mathcal{L}(x_i, y_i, z_i')$

        Update $\mathcal{M}_{\text{pre}}$ parameters via backpropagation

**return** $\mathcal{M}_{\text{pre}}$ as $\mathcal{M}_{\text{finetuned}}$

---

## 4    Experiment Setup

In this section, we detail the datasets employed for evaluating our plugin, along with the baseline pre-trained models and experimental configurations.

### 4.1    Datasets

To validate the performance of our method in text classification tasks, we selected a benchmark specifically focused on investigating spurious correlations in text classification, Shortcut Maze Zhou et al. (2024), as our evaluation platform, and chose three datasets from it as our experiment datasets.

This dataset achieves fine-grained exploration of spurious correlations by rigorously adjusting the parameter $\lambda$ to control the degree of association between data and labels, thereby regulating the strength of spurious correlations. The benchmark comprises three datasets: Yelp, Go Emotions, and Beer, and investigates three distinct types of spurious correlations—Occurrence, Style, and Concept—by constructing subsets within each dataset. Evaluating model performance on this benchmark enables a comprehensive assessment of the model's robustness against various types of spurious correlations. The following is information regarding the datasets we have selected from this benchmark.

| Dataset Name | Class Numbers | Scale | Shortcut Type |
|---|---|---|---|
| Yelp | 5 | 8,900/10,000/10,000 | Occurrence |
| Go Emotion | 4 | 1625/685/685 | Style |
| Beer | 4 | 2,000/2,000/2,000 | Concept |

Table 1: **Benchmark Dataset Characteristics.** In the 'Scale' column, the first number represents the size of the **training** set, the second number denotes the size of the **conventional test** set, and the third number indicates the size of the **anti-test** set.

### 4.2    Baselines

To benchmark the robustness of our MSFT plugin against spurious correlations, we conducted comparative experiments employing BERT Devlin et al. (2019) as a common backbone architecture, against several prevailing mitigation methods. These included DRO Sagawa et al. (2019), DFR Kirichenko et al. (2022), LLR LaBonte et al. (2023), and the family of NFL Chew et al. (2023) loss functions. Comparative analysis with these strong baselines effectively validates the performance of our MSFT approach.

Furthermore, to more broadly validate the applicability of MSFT, we integrated it into multiple pre-trained models and compared the performance gains against their original counterparts. This evaluation encompassed three Masked Language Models (MLMs)—DistilBERT Sanh et al. (2019), RoBERTa Liu et al. (2019), and DeBERTa He et al. (2020)—as well as two Large Language Models (LLMs) of different scales: GPT-2 and GPT-2 Medium Radford et al. (2019).

### 4.3    Metrics, Hyperparameters and Setup for Experiment

The Shortcut Maze dataset employs three metrics to evaluate a model's robustness against spurious correlations: the accuracy on the conventional test set and the anti-test set, as well as the difference (denoted as $\triangle$) between these two accuracies. Since the construction of the anti-test set is designed such that if a model over-relies on spurious correlations, its performance on the anti-test set will significantly degrade, a smaller value of $\triangle$ indicates better robustness to spurious correlations.

To identify the optimal value for the stretch ratio, we performed a grid search over MSFT, considering stretch ratios of 0.1, 0.5, 2, 10, 100, and 1000 as candidate values (representing different orders of magnitude).

Finally, for each model, we conducted ten training runs and took the average of the last five runs as the final result for a single training instance. To ensure the statistical reliability of our results, we performed multiple training runs for statistical significance.

| Model | Beer | | | Yelp | | | Emotion | | |
|---|---|---|---|---|---|---|---|---|---|
| | ACC(%) | $ACC_{anti}$(%) | $\triangle$(%) | ACC(%) | $ACC_{anti}$(%) | $\triangle$(%) | ACC(%) | $ACC_{anti}$(%) | $\triangle$(%) |
| NFL-CO | $86.74_{\pm0.76}$ | $63.10_{\pm3.78}$ | 23.64 | $63.12_{\pm1.03}$ | $41.82_{\pm1.96}$ | 21.30 | $73.65_{\pm1.28}$ | $16.62_{\pm1.04}$ | 57.03 |
| NFL-CP | $59.40_{\pm0.81}$ | $44.04_{\pm3.70}$ | ~~15.36~~ | $55.83_{\pm3.64}$ | $35.45_{\pm3.89}$ | ~~20.38~~ | $74.51_{\pm2.72}$ | $6.28_{\pm1.08}$ | 68.23 |
| DRO | $37.51_{\pm21.70}$ | $43.48_{\pm13.43}$ | ~~-5.97~~ | $44.88_{\pm20.32}$ | $28.18_{\pm6.74}$ | ~~18.70~~ | $64.84_{\pm6.58}$ | $11.97_{\pm1.88}$ | ~~52.87~~ |
| DFR | $54.07_{\pm0.18}$ | $42.56_{\pm0.87}$ | ~~11.49~~ | $49.18_{\pm0.49}$ | $\mathbf{45.93_{\pm0.22}}$ | 3.25 | $69.33_{\pm0.80}$ | $11.65_{\pm0.56}$ | ~~57.68~~ |
| LLR | $78.53_{\pm3.56}$ | $49.22_{\pm8.53}$ | ~~29.31~~ | $60.60_{\pm0.40}$ | $42.79_{\pm2.53}$ | **17.81** | $\mathbf{78.92_{\pm2.05}}$ | $14.39_{\pm3.37}$ | 64.53 |
| MSFT(Ours) | $\mathbf{88.40_{\pm1.04}}$ | $\mathbf{67.54_{\pm2.72}}$ | 20.86 | $\mathbf{64.13_{\pm0.20}}$ | $39.18_{\pm1.06}$ | 24.95 | $74.06_{\pm1.48}$ | $\mathbf{18.98_{\pm2.27}}$ | **55.08** |

Table 2: **Performance Results of MSFT versus Baselines on ACC and $ACC_{anti}$ in Shoutcuts Maze.** OSR rates of MSFT on the three datasets are 0.5, 2, and 0.1, respectively. The best performing model is highlighted in bold. F1 and $F1_{anti}$ results are presented in Appendix Table 17.

| Model | Beer | | | Yelp | | | Emotion | | |
|---|---|---|---|---|---|---|---|---|---|
| | ACC(%) | $ACC_{anti}$(%) | $\triangle$(%) | ACC(%) | $ACC_{anti}$(%) | $\triangle$(%) | ACC(%) | $ACC_{anti}$(%) | $\triangle$(%) |
| DeBERTa | **92.13** | 65.16 | 26.97 | 69.30 | 40.80 | 28.50 | 70.56 | **18.42** | **52.14** |
| DeBERTa+MSFT | 91.66 | **71.92** | **19.74** | **69.43** | **51.09** | **18.33** | **79.01** | 15.56 | 63.45 |
| DistilBERT | **89.91** | **65.04** | 24.87 | 63.14 | **37.77** | 25.37 | **75.59** | **17.28** | 58.31 |
| DistilBERT+MSFT | 85.50 | 62.06 | **23.44** | **63.22** | 38.46 | **24.75** | 74.28 | 16.61 | **57.66** |
| RoBERTa | **91.83** | **67.63** | **24.20** | **66.60** | 44.84 | 21.76 | 72.99 | **15.56** | 57.43 |
| RoBERTa+MSFT | 91.16 | 60.50 | 30.66 | 66.53 | **51.06** | **15.48** | **77.49** | 13.90 | 63.59 |
| GPT-2 | **90.29** | **83.96** | **6.33** | **64.69** | 40.32 | 24.37 | 68.90 | 14.54 | 54.36 |
| GPT-2+MSFT | 85.02 | 74.09 | 10.93 | 61.68 | **46.75** | **14.93** | **72.03** | **17.81** | **54.22** |
| GPT-2-medium | 90.93 | 83.77 | 7.16 | 66.09 | 43.39 | 22.70 | **74.45** | 14.33 | 60.12 |
| GPT-2-medium+MSFT | **91.58** | **86.24** | **5.34** | **66.83** | **47.17** | **19.66** | 72.55 | **16.38** | **56.18** |

Table 3: **Benchmarking MSFT (under OSR) against Multiple Vanilla Backbone Architectures**: A Comparative Study on Metrics ACC, $ACC_{anti}$, and $\triangle$. Dominant results in the comparative evaluation are denoted using bold formatting. F1 and $F1_{anti}$ results are presented in Appendix Table 18.

## 5 Results

In this chapter, we will present the experimental results in two parts: a comparative analysis with other methods addressing spurious correlations to demonstrate the effectiveness of MSFT, and a comparison with the backbone model to validate its generalizability.

### 5.1 Comparative Results with Other Spurious Correlation Solutions

We conducted experimental comparisons of all selected methods for addressing spurious correlations across three sub-datasets, following the evaluation metrics of the Shortcuts Maze dataset. The experimental results are presented in Tables 2. It is specifically noted that when the test set accuracy of a method is substantially lower than that of other methods, its corresponding Metric $\triangle$ is considered invalid. Therefore, comparisons based on Metric $\triangle$ are only performed under comparable levels of test set accuracy. Invalid metrics are marked with strikethrough in the tables and are excluded from the evaluation scope.

As can be observed from the table, among the three loss function approaches, only NFL-CO exhibits relatively robust performance across all three datasets, whereas both NFL-CP and the classical DRO method perform poorly. Specifically, the DRO loss function fails completely on both the Beer and Yelp datasets, with its performance lagging significantly behind other spurious correlation mitigation approaches. All three $\triangle$ metrics for DRO are invalid, indicating that the DRO loss function is ineffective in resolving the multiple spurious correlations present in the Shortcuts Maze dataset. Among the two training strategies evaluated, while DFR demonstrates superior performance compared to DRO, its predictive effectiveness remains suboptimal on most test sets. In contrast, LLR, an enhanced strategy derived from DFR, delivers notably robust performance and achieves SOTA results on the Yelp dataset among existing approaches for mitigating spurious correlations. In summary, within the context of performance on the Shortcuts Maze dataset, NFL-CO emerges as the

optimal choice among loss function methods, whereas LLR represents the most effective training strategy.

Our proposed plug-in, MSFT, achieves high performance across all three datasets, attaining SOTA levels on the Beer and Emotion datasets and comprehensively surpassing existing methods for addressing spurious correlations in text classification. To the best of our knowledge, MSFT represents the first plug-and-play approach for addressing spurious correlations in text classification that operates independently of specialized loss functions or training strategies, while achieving SOTA performance on the Shortcuts Maze benchmark.

## 5.2 COMPARATIVE RESULTS WITH THE BACKBONE MODEL

Table 3 presents the comparative results between MSFT and the vanilla backbone model on the Shortcuts Maze benchmark.

As shown in the table, MSFT achieves a significant reduction in Metric $\triangle$ across multiple pretrained models and datasets, demonstrating its effectiveness in mitigating the negative impact of spurious correlations during fine-tuning and enhancing model robustness. These results validate the generalizability of the MSFT plugin.

For MLM-style models, the MSFT technique not only facilitated an improvement in accuracy on general test sets but also achieved a reduction in $\triangle$ across multiple datasets. This indicates that for most MLM-style pre-trained models, MSFT can enhance robustness against various spurious correlations without compromising the model's predictive capabilities. We posit that this benefit stems from the plug-in nature of our approach. Since we do not alter the loss function of the backbone model, the mitigation of spurious correlations does not exert a global influence on the model's parameters. Consequently, this mitigation strategy avoids introducing other significant negative side effects to the model's learning process.

As for autoregressive language models, the performance of MSFT on GPT2-Medium was markedly superior to its performance on the smaller GPT model. We hypothesize that this discrepancy may be attributable to the inherent characteristics of autoregressive architectures. Autoregressive models with fewer parameters may lack the capacity to effectively adapt to samples transformed via the embedding layer stretching, leading to performance instability. In contrast, the larger parameter count of GPT2-Medium enables it to satisfactorily accommodate a wider variety of embedding layer stretching transformations, thereby achieving a more substantial enhancement in robustness against spurious correlations.

## 6 ABLATION STUDY

### 6.1 PROCESSING EXCESS SAMPLES

| SR | Beer | | | Yelp | | | Emotion | | |
|---|---|---|---|---|---|---|---|---|---|
| | ACC(%) | $\text{ACC}_{\text{anti}}$(%) | $\triangle$(%) | ACC(%) | $\text{ACC}_{\text{anti}}$(%) | $\triangle$(%) | ACC(%) | $\text{ACC}_{\text{anti}}$(%) | $\triangle$(%) |
| $\text{Drop}_{\text{Excess Samples}}$ | 75.18 | 58.07 | ~~17.11~~ | 61.95 | 38.46 | **23.49** | 65.54 | 17.57 | ~~47.97~~ |
| MSFT(Ours) | **88.40** | **67.54** | **20.86** | **64.13** | **39.18** | 24.95 | **74.06** | **18.98** | **55.08** |

Table 4: **Ablation Study on MSFT: Performance Comparison Between Training with Stretched Redundant Data and Training with Deleted Redundant Data.**

Following the computation of semantic centroids via the K-Means algorithm and subsequent clustering of dataset embeddings, MSFT identifies all excess samples. Building upon the concept of downsampling, we leverage these excess samples—which would typically be discarded in conventional downsampling—and iteratively incorporate them during training through a stretching mechanism to enhance robustness against spurious correlations. To validate the efficacy of this approach, we compare it against direct elimination of such excess samples, i.e., implementing standard downsampling directly atop our plugin-based classifier. The experimental outcomes are summarized in Table 4.

The experimental results indicate that directly discarding all excess samples identified by MSFT fails to yield significant accuracy improvements in model learning, which may be attributed to the inher-

ent design characteristics of the Shortcuts Maze dataset. This dataset incorporates multiple spurious correlations during its construction and introduces $Test_{anti}$ through adjusted masking parameters, making simple subgroup balancing insufficient for effectively mitigating spurious correlations. This scenario reflects the prevailing challenge in most practical contexts: the diverse nature of spurious correlations renders strategies based on single-dimensional subgroup partitioning and balancing inherently inefficient. In contrast, the stretched excess samples enable model learning along novel dimensions, thereby enhancing robustness against spurious correlations.

## 6.2 INVESTIGATION OF THE STRETCHING RATIO'S INFLUENCE ON MSFT

To investigate the effectiveness of the stretching ratio hyperparameter, this section first analyzes the performance of the BERT-based MSFT plugin across three datasets. We subsequently document the accuracy trends during training under varying stretching ratios, examining their dynamic impact on model learning to delineate the appropriate selection range for this parameter.

### 6.2.1 ANALYSIS OF MSFT PERFORMANCE UNDER VARYING STRETCHING RATIOS

| SR | Beer | | | Yelp | | | Emotion | | |
|---|---|---|---|---|---|---|---|---|---|
| | ACC(%) | $ACC_{anti}$(%) | $\triangle$(%) | ACC(%) | $ACC_{anti}$(%) | $\triangle$(%) | ACC(%) | $ACC_{anti}$(%) | $\triangle$(%) |
| 0.1 | $86.48_{\pm0.47}$ | $61.06_{\pm2.65}$ | 25.41 | $63.51_{\pm0.38}$ | $39.38_{\pm0.50}$ | 24.14 | $74.06_{\pm1.48}$ | $18.98_{\pm2.27}$ | 55.08 |
| 0.5 | $88.40_{\pm1.04}$ | $67.54_{\pm2.72}$ | 20.86 | $64.06_{\pm0.21}$ | $39.61_{\pm0.63}$ | 24.46 | $76.02_{\pm0.61}$ | $17.14_{\pm0.88}$ | 58.87 |
| 2 | $88.03_{\pm0.70}$ | $66.66_{\pm1.91}$ | 21.37 | $64.13_{\pm0.20}$ | $39.18_{\pm1.06}$ | 24.95 | $74.59_{\pm0.80}$ | $16.84_{\pm0.98}$ | 57.75 |
| 10 | $86.72_{\pm0.25}$ | $64.73_{\pm1.20}$ | 21.99 | $64.10_{\pm0.18}$ | $40.07_{\pm1.59}$ | 24.03 | $73.55_{\pm0.68}$ | $10.44_{\pm1.37}$ | 63.12 |
| 100 | $78.13_{\pm1.11}$ | $56.83_{\pm6.93}$ | 21.30 | $61.13_{\pm0.59}$ | $40.21_{\pm2.00}$ | 20.92 | $74.17_{\pm3.73}$ | $9.68_{\pm0.65}$ | 64.50 |
| 1000 | $80.30_{\pm1.63}$ | $52.35_{\pm5.14}$ | 27.95 | $58.26_{\pm2.52}$ | $40.58_{\pm2.43}$ | 17.67 | $76.54_{\pm1.35}$ | $9.00_{\pm1.60}$ | 67.54 |

Table 5: **ACC, ACC2 and $\triangle$ Performance of the Shortcuts Maze Model under Varying Stretching Ratios.** F1 and $F1_{anti}$ results are presented in Appendix Table 19.

Table 5 presents the performance of the BERT-based MSFT framework across three datasets under varying stretching ratios.

On the Beer dataset, performance is suboptimal at SR=0.1. As the stretching ratio increases, the model progressively enhances its robustness against spurious correlations, evidenced by a continuous decrease in $\triangle$. The results stabilize at SR values of 0.5, 2, and 10. However, when SR reaches magnitudes of 100 and 1000, predictive accuracy on both test sets declines significantly, accompanied by a rise in $\triangle$, indicating diminished robustness to spurious correlations. In contrast, the Yelp dataset maintains relatively stable performance across SR values of 0.1, 0.5, 2, 10, and 100. Only at SR=1000 do both accuracy and $\triangle$ deteriorate markedly. For the Emotion dataset, while accuracy on the standard test set remains largely consistent across multiple SR values, a sharp decline is observed on the $Test_{anti}$ when SR reaches 100.

These findings demonstrate that the stretching ratio is a critical hyperparameter for MSFT, with its sensitivity being highly dataset-dependent. Among the three datasets, Yelp exhibits lower sensitivity to SR variations, Beer shows higher sensitivity, and Emotion displays a notable divergence: its standard test set remains largely unaffected, whereas its $Test_{anti}$ is exceptionally sensitive. This indicates that selecting an appropriate SR profoundly influences the mitigation of diverse spurious correlations, and different types of spurious correlations exhibit varying degrees of responsiveness to SR adjustments.

### 6.2.2 ANALYSIS OF THE IMPACT OF DIFFERENT STRETCHING RATIOS ON THE LEARNING PROCESS

Figure 3 illustrates the line chart depicting the per-epoch learning variations and corresponding performance of MSFT based on BERT under different stretching ratios on the Yelp dataset. In the figure, the red curve represents the trend of prediction accuracy for non-excess data during the training process, while the blue curve indicates the trend of prediction accuracy for excess data throughout the training procedure.

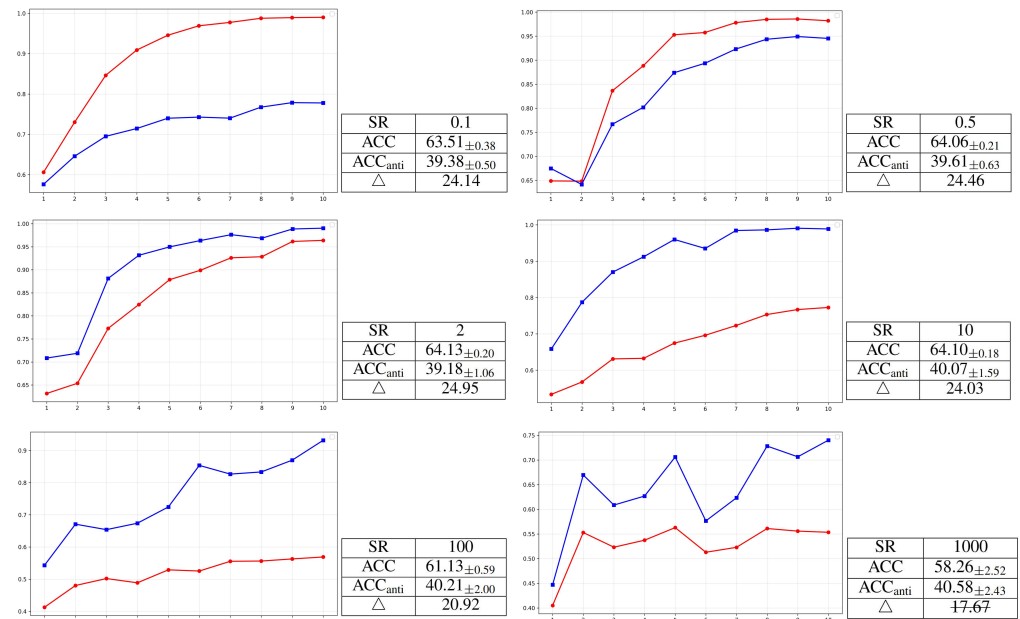

Figure 3: **Line Graph Depicting the Impact of Various Stretching Ratios on Model Learning Trajectory and Resultant Performance on the Yelp Dataset.**

As can be observed from the figure regarding the Yelp dataset, the model's learning process is substantially influenced by variations in the stretching ratio.

When the SR is 0.1, excess samples become highly concentrated near the semantic centroid, making it difficult for the model to effectively learn and distinguish these samples, thereby resulting in notably low prediction accuracy for such samples. As the SR increases to 0.5, the aggregation degree of excess samples decreases, leading to relatively clearer decision boundaries between samples and a corresponding recovery in the accuracy of excess samples. At an SR of 2, an optimal state is reached, where both excess and non-excess samples are perfectly fitted at different training stages, ultimately converging to high accuracy rates and achieving feature disentanglement. However, when the SR further increases to 10, 100, or even 1000, excessive stretching of the embedding layer impedes model fitting on the Yelp dataset. The non-excess samples become relatively clustered and hard to distinguish, ultimately leading to unsatisfactory prediction performance on non-excess samples, a significant decline in conventional test accuracy (58.26%), and the invalidation of metric $\triangle$.

In summary, identifying an appropriate SR for the dataset can effectively enhance the performance of the MSFT framework. Since this plugin operates exclusively on the embedding layer and its stretching characteristics—unlike loss functions—do not extensively alter parameters during back-propagation, it effectively enhances robustness against spurious correlations without significantly compromising predictive accuracy.

## 7    CONCLUSION

In this paper, we introduce MSFT, a novel text classification plugin that mitigates spurious corre-lations via semantic centroid identification and tensor stretching. By automatically detecting latent clusters in the embedding space, MSFT identifies and processes excess samples driving spurious correlations—eliminating the need for prior spurious attribute knowledge or additional data anno-tations. Notably, MSFT outperforms existing methods for mitigating spurious correlations in text classification on the Shortcuts Maze benchmark, while demonstrating broad compatibility with a diverse range of pre-trained models—all of which have been widely adopted in the field of text clas-sification. These results collectively demonstrate the practical viability and broad compatibility of the MSFT framework.

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

# A    APPENDIX

## A.1    ETHICAL STATEMENT

We exclusively employ large language models for polishing the English text of academic papers, with no additional applications.

| Notation | Description | Notation | Description |
|---|---|---|---|
| $\mathcal{A}$ | Set of spurious attributes | $\mathcal{G}$ | Set of group labels $\mathcal{Y} \times \mathcal{A}$ |
| $C_i$ | Cluster group $i$ | $\mathcal{L}_{C_i}^+$ | Positive labels in $C_i$ |
| $\mathcal{L}_{C_i}^-$ | Negative labels in $C_i$ | $\min_{y \in Y}$ | Minority class size |
| $\mathbf{s}$ | Semantic centroid | $\mathcal{C}$ | Cluster centers $\{c_p\}_{p=1}^{m^*}$ |
| $m^*$ | Optimal cluster count | $z_i$ | Embedding tensors from model |
| $z_i'$ | Modified embedding | $\alpha$ | Stretching ratio |

Table 6: **Chapter 3 and Appendix A.2 Mathematical Notation Summary.**

## A.2    DEFINITION OF SPURIOUS CORRELATION

Let $\mathcal{D}_{\text{tr}} = \{(\mathbf{x}_i, y_i)\}_{i=1}^n$ be a training dataset where $\mathbf{x}_i \in \mathcal{X}$ denotes the set of all possible inputs and $y_i \in \mathcal{Y}$ the class label ($\mathcal{Y}$ contains $K$ classes). Each $\mathbf{x}_i$ is associated with a *non-predictive* spurious attribute $a_i \in \mathcal{A}$. A spurious correlation $\langle y, a \rangle$ describes an association between class $y \in \mathcal{Y}$ and attribute $a \in \mathcal{A}$, where $\phi : \mathcal{A} \mapsto \mathcal{Y}^{K'}$ defines a one-to-many mapping conditioned on $\mathcal{D}_{\text{tr}}$ ($1 < K' \leq K$). Data with correlation $\langle y, a \rangle$ are annotated with group label $g = (y, a) \in \mathcal{G} := \mathcal{Y} \times \mathcal{A}$.

For example, in the context of conceptual spurious correlations, let $\mathcal{A}$ denote conceptual spurious correlations, with $a_i$ representing distinct concepts (e.g., color, size, etc.). When the training set contains a substantial number of color-associated instances predominantly labeled as positive, the model may infer a spurious positive correlation between color and positive labels, formalized as $\langle y, a \rangle$. However, if a test instance contains color-related features but is labeled as negative (denoted as $\langle y', a \rangle$), the model will produce incorrect predictions due to this learned spurious association.

## A.3    DETAILED INTRODUCTION AND RELATED WORK

### A.3.1    DETAILED INTRODUCTION

Early research in data augmentation for spurious correlations in text classification evolved from addressing biases in small-scale datasets Torralba & Efros (2011) to hypothesis-driven learning frameworks Poliak et al. (2018); Gururangan et al. (2018) and QA/paragraph-based training approaches Kaushik & Lipton (2018); Feng et al. (2018). Recent advancements have explored counterfactual example generation Volodin et al. (2020); Zeng et al. (2020); Hamman et al. (2023), alongside annotation-based bias correction Srivastava et al. (2020), semantic perturbation techniques Puli et al. (2022), and unstable feature detection methods Wu et al. (2023). Nevertheless, data augmentation

primarily perturbs low-level features, struggling to model semantically consistent high-level variations. Meanwhile, counterfactual example-based methods are limited to datasets with binary label attributes (e.g., sentiment classification) and remain ill-suited for topic classification and most other text classification tasks.

Current learning strategies for mitigating spurious correlations primarily operate through two avenues: loss optimization techniques (e.g., group distribution regularization Sagawa et al. (2020); Levy et al. (2020) and Logit loss minimization Liu et al. (2022)) and ensemble frameworks (encompassing single-stage Kim et al. (2022) and multi-stage approaches Lee et al. (2022); Nam et al. (2020)). Other methodologies include spurious correlation removal techniques Liu et al. (2021); Yenamandra et al. (2023); Yang et al. (2024), fine-tuning approaches Asgari et al. (2022); Ludan et al. (2023); LaBonte et al. (2023), and adversarial learning paradigms He et al. (2023). Nevertheless, these methods are constrained by limitations such as hyperparameter sensitivity and architecture dependency.

Within the domain of representation learning, research efforts are broadly categorized into three analytical levels: token-level Wang et al. (2021); Tang et al. (2023); Chew et al. (2023), sentence/document-level Borah et al. (2023), and concept-level Principe et al. (2024); Zhou et al. (2023). Complementary research threads further include feature disentanglement techniques Yang et al. (2022); Lee et al. (2022) and invariant learning methodologies Eastwood et al. (2023); Chen et al. (2022).

### A.3.2 DETAILED RELATED WORK

The 'identify-then-mitigate' paradigm for addressing spurious correlations has gained increasing traction in recent years. Liu et al. (2021) introduces a straightforward yet effective approach: first identifying challenging samples, then retraining the model with adjusted sample weights to amplify the influence of these samples during training, thereby mitigating spurious correlations. Yenamandra et al. (2023) leverages empirical risk minimization under simplified bias assumptions to strengthen target correlations, followed by correlation-aware slicing and mixture modeling to rectify underperforming data slices exhibiting divergent correlations. Yang et al. (2024) detects majority groups prone to spurious correlations early in training, then applies importance sampling to rebalance group distributions and diminish their adverse effects. A common limitation of these approaches, however, is their neglect of the dataset's global structural properties.

The fine-tuning stage serves as a critical intervention point for mitigating the impact of spurious correlations on model performance. Asgari et al. (2022) encourages the model to attend to a broader set of input variables by aligning their representations to a shared target, fostering a more comprehensive data understanding. Ludan et al. (2023) involves pre-training the model on artificially constructed datasets with diverse spurious cues, then evaluating on cue-free datasets—this methodology underscores the necessity of mitigating spurious correlations during fine-tuning. LaBonte et al. (2023) fine-tunes only the final layer using small, balanced, and unlabeled datasets, achieving performance comparable to DFR. Nevertheless, these works rarely explore the embedding layer, despite its more direct association with spurious correlations.

The emerging field of concept-level spurious correlation research has yielded novel insights for mitigating spurious associations. Principe et al. (2024) applied K-means clustering to perform concept-level reclassification of words in text segments; subsequent application of these classification results to reorganize text segments led to measurable improvements in model performance, thereby validating the importance of concept-level representations. Through manual annotation of multiple datasets, Zhou et al. (2023) confirmed the widespread impact of concept-level spurious correlations and proved downsampling as a lightweight mitigation strategy. However, the former introduces additional modules that increase model complexity, whereas the latter depends on pre-annotated concepts, which limits its generalizability.

| SR | Beer | | | Yelp | | | Emotion | | |
|---|---|---|---|---|---|---|---|---|---|
| | ACC(%) | ACC$_{anti}$(%) | $\triangle$(%) | ACC(%) | ACC$_{anti}$(%) | $\triangle$(%) | ACC(%) | ACC$_{anti}$(%) | $\triangle$(%) |
| 0.1 | 91.79 | 67.46 | 24.33 | 69.21 | 54.02 | 15.19 | 67.45 | 14.16 | 53.28 |
| 0.5 | 91.66 | 71.92 | 19.74 | 69.33 | 48.26 | 21.07 | 73.61 | 14.19 | 59.42 |
| 2 | 92.82 | 64.31 | 28.51 | 69.43 | 51.09 | 18.33 | 79.01 | 15.56 | 63.45 |
| 10 | 90.11 | 62.58 | 27.53 | 68.29 | 47.18 | 21.11 | 78.34 | 14.80 | 63.53 |
| 100 | 78.58 | 60.48 | 18.10 | 66.92 | 59.04 | 7.88 | 77.49 | 10.89 | 66.60 |
| 1000 | 79.58 | 55.52 | 24.06 | 65.01 | 51.91 | 13.11 | 75.91 | 16.47 | 59.45 |

Table 7: Comparative Analysis of ACC, ACC$_{anti}$ and $\triangle$ Metrics for MSFT across Different Stretch Ratios Utilizing DeBERT Backbone Architecture.

| SR | Beer | | | Yelp | | | Emotion | | |
|---|---|---|---|---|---|---|---|---|---|
| | ACC(%) | ACC$_{anti}$(%) | $\triangle$(%) | ACC(%) | ACC$_{anti}$(%) | $\triangle$(%) | ACC(%) | ACC$_{anti}$(%) | $\triangle$(%) |
| 0.1 | 85.50 | 62.06 | 23.44 | 61.92 | 39.57 | 22.35 | 74.28 | 16.61 | 57.66 |
| 0.5 | 87.10 | 58.83 | 28.27 | 63.22 | 38.46 | 24.75 | 75.80 | 16.61 | 59.18 |
| 2 | 86.36 | 56.12 | 30.24 | 63.13 | 38.13 | 25.00 | 76.26 | 15.04 | 61.23 |
| 10 | 85.59 | 50.68 | 34.91 | 62.99 | 39.46 | 23.53 | 74.22 | 13.64 | 60.58 |
| 100 | 74.25 | 47.67 | 26.58 | 61.60 | 39.33 | 22.27 | 72.29 | 11.42 | 60.88 |
| 1000 | 70.56 | 44.45 | 26.11 | 59.61 | 39.84 | 19.77 | 73.55 | 11.04 | 62.51 |

Table 8: Comparative Analysis of ACC, ACC$_{anti}$ and $\triangle$ Metrics for MSFT across Different Stretch Ratios Utilizing DistilBERT Backbone Architecture.

| SR | Beer | | | Yelp | | | Emotion | | |
|---|---|---|---|---|---|---|---|---|---|
| | ACC(%) | ACC$_{anti}$(%) | $\triangle$(%) | ACC(%) | ACC$_{anti}$(%) | $\triangle$(%) | ACC(%) | ACC$_{anti}$(%) | $\triangle$(%) |
| 0.1 | 89.65 | 59.63 | 30.02 | 66.28 | 44.70 | 21.58 | 69.69 | 17.93 | 51.77 |
| 0.5 | 90.23 | 62.74 | 27.49 | 66.56 | 45.24 | 21.32 | 73.99 | 16.18 | 57.81 |
| 2 | 91.16 | 60.50 | 30.66 | 66.89 | 44.18 | 22.71 | 73.58 | 15.88 | 57.69 |
| 10 | 88.51 | 60.70 | 27.81 | 66.53 | 51.06 | 15.48 | 77.49 | 13.90 | 63.59 |
| 100 | 81.25 | 53.65 | 27.60 | 64.60 | 49.10 | 15.51 | 73.40 | 10.57 | 62.83 |
| 1000 | 81.36 | 50.94 | 30.42 | 62.78 | 45.09 | 17.69 | 73.58 | 9.43 | 64.15 |

Table 9: Comparative Analysis of ACC, ACC$_{anti}$ and $\triangle$ Metrics for MSFT across Different Stretch Ratios Utilizing RoBERT Backbone Architecture.

| SR | Beer | | | Yelp | | | Emotion | | |
|---|---|---|---|---|---|---|---|---|---|
| | ACC(%) | ACC$_{anti}$(%) | $\triangle$(%) | ACC(%) | ACC$_{anti}$(%) | $\triangle$(%) | ACC(%) | ACC$_{anti}$(%) | $\triangle$(%) |
| 0.1 | 80.75 | 70.91 | 9.84 | 63.94 | 40.23 | 23.72 | 69.61 | 13.37 | 56.23 |
| 0.5 | 85.02 | 74.09 | 10.93 | 64.16 | 41.07 | 23.09 | 72.64 | 14.80 | 57.84 |
| 2 | 84.24 | 71.27 | 12.97 | 65.22 | 40.49 | 24.73 | 70.60 | 12.12 | 58.48 |
| 10 | 79.03 | 57.10 | 21.93 | 63.96 | 42.01 | 21.95 | 72.03 | 17.81 | 54.22 |
| 100 | 63.83 | 36.59 | 27.24 | 61.68 | 46.75 | 14.93 | 72.73 | 14.72 | 58.01 |
| 1000 | 60.82 | 34.71 | 26.11 | 61.24 | 38.97 | 22.27 | 66.69 | 16.06 | 50.63 |

Table 10: Comparative Analysis of ACC, ACC$_{anti}$ and $\triangle$ Metrics for MSFT across Different Stretch Ratios Utilizing GPT2 Backbone Architecture.

| SR | Beer | | | Yelp | | | Emotion | | |
|----|------|------|------|------|------|------|---------|------|------|
| | ACC(%) | ACC$_{anti}$(%) | $\triangle$(%) | ACC(%) | ACC$_{anti}$(%) | $\triangle$(%) | ACC(%) | ACC$_{anti}$(%) | $\triangle$(%) |
| 0.1 | 92.10 | 84.46 | 7.64 | 66.83 | 47.17 | 19.66 | 78.16 | 12.03 | 66.13 |
| 0.5 | 91.60 | 84.19 | 7.41 | 66.08 | 29.40 | 36.68 | 70.80 | 14.28 | 56.53 |
| 2 | 92.52 | 85.70 | 6.82 | 66.07 | 21.97 | 44.10 | 72.55 | 16.38 | 56.18 |
| 10 | 91.58 | 86.24 | 5.34 | 66.44 | 40.77 | 25.67 | 69.55 | 16.29 | 53.26 |
| 100 | 84.11 | 75.36 | 8.75 | 23.84 | 46.28 | -22.45 | 67.59 | 16.88 | 50.72 |
| 1000 | 79.35 | 81.73 | -2.38 | 25.75 | 42.81 | -17.06 | 67.77 | 16.93 | 50.83 |

Table 11: Comparative Analysis of ACC, ACC$_{anti}$ and $\triangle$ Metrics for MSFT across Different Stretch Ratios Utilizing GPT2-Medium Backbone Architecture.

| SR | Beer | | Yelp | | Emotion | |
|----|------|------|------|------|---------|------|
| | F1(%) | F1$_{anti}$(%) | F1(%) | F1$_{anti}$(%) | F1(%) | F1$_{anti}$(%) |
| 0.1 | 91.70 | 67.51 | 69.17 | 53.69 | 70.33 | 7.95 |
| 0.5 | 91.60 | 71.80 | 69.15 | 45.85 | 76.02 | 8.67 |
| 2 | 92.77 | 64.54 | 69.15 | 49.04 | 80.09 | 11.79 |
| 10 | 89.89 | 62.70 | 68.12 | 44.29 | 79.50 | 10.64 |
| 100 | 75.97 | 55.44 | 66.60 | 59.30 | 78.73 | 9.09 |
| 1000 | 77.30 | 53.59 | 65.25 | 51.61 | 77.44 | 15.57 |

Table 12: Comparative Analysis of F1 and F1$_{anti}$ Metrics for MSFT across Different Stretch Ratios Utilizing DeBERT Backbone Architecture.

| SR | Beer | | Yelp | | Emotion | |
|----|------|------|------|------|---------|------|
| | F1(%) | F1$_{anti}$(%) | F1(%) | F1$_{anti}$(%) | F1(%) | F1$_{anti}$(%) |
| 0.1 | 85.41 | 62.22 | 60.99 | 38.09 | 76.24 | 10.56 |
| 0.5 | 86.98 | 58.81 | 62.82 | 35.28 | 76.67 | 10.26 |
| 2 | 86.26 | 56.18 | 63.05 | 34.44 | 77.95 | 8.06 |
| 10 | 85.48 | 50.61 | 63.18 | 35.86 | 76.32 | 7.55 |
| 100 | 73.88 | 46.62 | 61.57 | 36.39 | 74.17 | 7.14 |
| 1000 | 69.25 | 43.78 | 60.14 | 37.78 | 74.93 | 8.64 |

Table 13: Comparative Analysis of F1 and F1$_{anti}$ Metrics for MSFT across Different Stretch Ratios Utilizing DistilBERT Backbone Architecture.

| SR | Beer | | Yelp | | Emotion | |
|---|---|---|---|---|---|---|
| | F1(%) | $F1_{anti}$(%) | F1(%) | $F1_{anti}$(%) | F1(%) | $F1_{anti}$(%) |
| 0.1 | 89.57 | 59.91 | 66.09 | 42.12 | 72.87 | 12.75 |
| 0.5 | 90.18 | 62.51 | 66.22 | 42.18 | 76.25 | 8.60 |
| 2 | 91.07 | 60.33 | 67.01 | 40.41 | 76.00 | 9.68 |
| 10 | 88.36 | 60.40 | 66.23 | 50.14 | 79.12 | 7.90 |
| 100 | 80.21 | 51.76 | 64.41 | 48.24 | 75.65 | 6.09 |
| 1000 | 79.36 | 47.05 | 62.31 | 40.37 | 75.74 | 9.70 |

Table 14: Comparative Analysis of F1 and $F1_{anti}$ Metrics for MSFT across Different Stretch Ratios Utilizing RoBERT Backbone Architecture.

| SR | Beer | | Yelp | | Emotion | |
|---|---|---|---|---|---|---|
| | F1(%) | $F1_{anti}$(%) | F1(%) | $F1_{anti}$(%) | F1(%) | $F1_{anti}$(%) |
| 0.1 | 80.47 | 70.57 | 63.22 | 36.38 | 72.56 | 8.27 |
| 0.5 | 84.86 | 73.97 | 63.68 | 36.80 | 74.66 | 8.86 |
| 2 | 83.90 | 71.24 | 64.73 | 35.46 | 73.52 | 7.35 |
| 10 | 77.56 | 56.10 | 63.83 | 39.08 | 73.82 | 15.91 |
| 100 | 60.63 | 32.75 | 60.10 | 46.85 | 74.55 | 10.63 |
| 1000 | 55.82 | 30.63 | 60.74 | 37.75 | 70.47 | 14.77 |

Table 15: Comparative Analysis of F1 and $F1_{anti}$ Metrics for MSFT across Different Stretch Ratios Utilizing GPT2 Backbone Architecture.

| SR | Beer | | Yelp | | Emotion | |
|---|---|---|---|---|---|---|
| | F1(%) | $F1_{anti}$(%) | F1(%) | $F1_{anti}$(%) | F1(%) | $F1_{anti}$(%) |
| 0.1 | 92.06 | 84.28 | 66.96 | 45.31 | 79.35 | 8.28 |
| 0.5 | 91.50 | 83.95 | 66.23 | 21.97 | 73.90 | 8.87 |
| 2 | 92.51 | 85.63 | 65.79 | 15.07 | 74.94 | 11.23 |
| 10 | 91.53 | 86.11 | 66.14 | 36.14 | 72.55 | 10.44 |
| 100 | 83.64 | 74.22 | 17.29 | 44.15 | 70.31 | 11.60 |
| 1000 | 78.20 | 80.70 | 19.26 | 38.52 | 70.39 | 15.85 |

Table 16: Comparative Analysis of F1 and $F1_{anti}$ Metrics for MSFT across Different Stretch Ratios Utilizing GPT2-Medium Backbone Architecture.

| Model | Beer | | Yelp | | Emotion | |
|---|---|---|---|---|---|---|
| | F1(%) | F1$_{anti}$(%) | F1(%) | F1$_{anti}$(%) | F1(%) | F1$_{anti}$(%) |
| NFL-CO | $86.68_{\pm0.76}$ | $63.33_{\pm3.78}$ | $63.09_{\pm1.05}$ | $39.58_{\pm2.99}$ | $76.04_{\pm1.06}$ | $11.90_{\pm1.60}$ |
| NFL-CP | $55.49_{\pm1.91}$ | $41.28_{\pm4.04}$ | $52.49_{\pm4.68}$ | $33.85_{\pm5.15}$ | $73.28_{\pm1.25}$ | $7.10_{\pm2.35}$ |
| DRO | $27.04_{\pm27.04}$ | $36.44_{\pm19.66}$ | $39.38_{\pm26.71}$ | $18.53_{\pm9.77}$ | $66.50_{\pm6.73}$ | $9.50_{\pm1.24}$ |
| DFR | $53.69_{\pm0.21}$ | $42.42_{\pm0.96}$ | $48.72_{\pm0.43}$ | $\mathbf{45.91}_{\pm0.19}$ | $72.13_{\pm0.64}$ | $11.13_{\pm0.58}$ |
| LLR | $78.42_{\pm3.65}$ | $48.76_{\pm9.24}$ | $59.83_{\pm0.38}$ | $41.52_{\pm2.95}$ | $\mathbf{78.85}_{\pm1.27}$ | $9.19_{\pm3.29}$ |
| MSFT(Ours) | $\mathbf{88.33}_{\pm1.03}$ | $\mathbf{67.73}_{\pm2.72}$ | $\mathbf{63.90}_{\pm0.18}$ | $35.28_{\pm1.70}$ | $76.16_{\pm1.27}$ | $\mathbf{14.69}_{\pm3.57}$ |

Table 17: **Performance Results of MSFT versus Baselines on F1 and F1$_{anti}$ in Shoutcuts Maze.** OSR rates of MSFT on the three datasets are 0.5, 2, and 0.1, respectively. The best performing model is highlighted in bold.

| Model | Beer | | Yelp | | Emotion | |
|---|---|---|---|---|---|---|
| | F1(%) | F1$_{anti}$(%) | F1(%) | F1$_{anti}$(%) | F1(%) | F1$_{anti}$(%) |
| DeBERTa | **92.05** | 65.27 | 69.11 | 33.48 | 73.53 | **15.47** |
| DeBERTa+MSFT | 91.60 | **67.51** | **69.15** | **49.04** | **80.09** | 11.79 |
| DistilBERT | **89.82** | **65.20** | **63.09** | 33.98 | **77.72** | **12.13** |
| DistilBERT+MSFT | 85.41 | 62.22 | 62.82 | **35.28** | 76.24 | 10.56 |
| RoBERTa | **91.75** | **67.63** | **66.59** | 41.56 | 75.67 | **10.20** |
| RoBERTa+MSFT | 91.07 | 60.33 | 66.23 | **50.14** | **79.12** | 7.90 |
| GPT-2 | **90.20** | **83.95** | **64.37** | 36.32 | 72.27 | 8.68 |
| GPT-2+MSFT | 84.86 | 73.97 | 60.10 | **46.85** | **73.82** | **15.91** |
| GPT-2-medium | 90.84 | 83.59 | 66.10 | 38.99 | **76.58** | 10.82 |
| GPT-2-medium+MSFT | **91.53** | **86.11** | **66.96** | **45.31** | 74.94 | **11.23** |

Table 18: **Benchmarking MSFT (under OSR) against Multiple Vanilla Backbone Architectures**: A Comparative Study on Metrics F1 and F1$_{anti}$. Dominant results in the comparative evaluation are denoted using bold formatting.

| SR | Beer | | Yelp | | Emotion | |
|---|---|---|---|---|---|---|
| | F1(%) | F1$_{anti}$(%) | F1(%) | F1$_{anti}$(%) | F1(%) | F1$_{anti}$(%) |
| 0.1 | $86.37_{\pm0.48}$ | $61.18_{\pm2.71}$ | $62.68_{\pm0.41}$ | $37.14_{\pm0.94}$ | $76.16_{\pm1.27}$ | $14.69_{\pm3.57}$ |
| 0.5 | $88.33_{\pm1.03}$ | $67.73_{\pm2.72}$ | $63.75_{\pm0.29}$ | $36.13_{\pm1.10}$ | $77.87_{\pm0.41}$ | $11.69_{\pm1.78}$ |
| 2 | $87.94_{\pm0.71}$ | $66.78_{\pm1.85}$ | $63.90_{\pm0.18}$ | $35.28_{\pm1.70}$ | $76.88_{\pm0.67}$ | $11.60_{\pm1.28}$ |
| 10 | $86.58_{\pm0.31}$ | $64.82_{\pm1.18}$ | $64.07_{\pm0.11}$ | $36.42_{\pm2.73}$ | $75.75_{\pm0.52}$ | $7.45_{\pm1.03}$ |
| 100 | $77.26_{\pm1.15}$ | $56.11_{\pm7.03}$ | $60.81_{\pm1.07}$ | $35.70_{\pm4.01}$ | $75.25_{\pm2.43}$ | $7.22_{\pm0.95}$ |
| 1000 | $80.06_{\pm1.66}$ | $51.53_{\pm5.47}$ | $56.03_{\pm3.22}$ | $36.32_{\pm4.54}$ | $76.54_{\pm0.45}$ | $8.34_{\pm2.36}$ |

Table 19: **F1 and F1$_{anti}$ Performance of the Shortcuts Maze Model under Varying Stretching Ratios.**

