# OpenReview forum: "MSFT: Mitigating Spurious Correlations in Text Classification via Feature Induction in Embedding Layers and Tensor Stretching"
_ICLR.cc/2026/Conference — Submitted to ICLR 2026_

### Official Review · Reviewer_GikN · 2025-10-22

**Soundness:** 2
**Presentation:** 2
**Contribution:** 2
**Rating:** 2
**Confidence:** 4

**Summary:**

This paper proposes a lightweight plugin, MSFT, that mitigates spurious correlations in text classification by manipulating embedding representations. It formalizes a “semantic centroid”, representing the global semantic information of a dataset, and introduces a tensor distance stretching mechanism in embedding space to weaken overrepresented, spurious associations. The authors claim MSFT can be seamlessly integrated into both BERT-style masked language models and autoregressive LLMs, achieving or surpassing state-of-the-art performance in mitigating spurious correlations.

**Strengths:**

(1) MSFT shows clear and repeatable gains in worst-group accuracy and overall accuracy across datasets and models.

(2) MSFT works effectively on both BERT-style and GPT-type models without modifying architecture.

(3) MSFT functions as a plug-in with only one main hyperparameter, making it simple and computationally cheap.

**Weaknesses:**

(1) The research gap is not clearly articulated. The introduction briefly mentions existing methods’ complexity and hyperparameter sensitivity but fails to explain why a new method is necessary, or how MSFT differs fundamentally from prior representation-level debiasing or centroid-based regularization approaches.

(2) The intuition -- that contracting embeddings toward a semantic centroid mitigates spurious correlation -- is presented without theoretical grounding or sufficient empirical justification.

(3) The definition of “excess samples” in Section 3.2.1 is ambiguous. It is unclear whether count(y) refers to class frequency within each cluster Ci or across the entire dataset.

(4) The notion that “excess samples are high-risk instances” is asserted without empirical evidence -- no visualization, distributional analysis, or ablation is provided to support that these samples indeed drive spurious correlations.

(5) The term “semantic centroid” is overclaimed as a “rigorously proven” construct. Its validation is purely empirical (via cosine similarity comparisons) and lacks mathematical rigor.

(6) Only one baseline family (NFL-CO/NFL-CP) is compared. The lack of broader baselines (e.g., GroupDRO, IRM, DFR, JTT) limits the credibility of “state-of-the-art” claims.

(7) The datasets used (Amazon, IMDB, Yahoo!) lack explicit spurious feature annotations. Hence, it is unclear what constitutes “spurious correlations” in these benchmarks.

(8) The evaluation metric (worst-group accuracy) is inadequately defined -- there is no explanation of how groups are constructed in datasets without subgroup labels, or why W-ACC can effectively demonstrate robustness to spurious correlation.

(9) Need a more detailed README file explaining how to use the shared code.

**Questions:**

(1) Clarify how count(y) is computed for “excess samples”.

(2) Explain why excess samples are considered high-risk.

(3) Specify which embedding layer is used for centroid computation and stretching.

(4) State whether centroids/clusters are computed once on training data or updated dynamically during training.

(5) Define what spurious features exist in the chosen datasets.

(6) Explain how subgroups for worst-group accuracy (W-ACC) are defined in datasets lacking group annotations. And justify why W-ACC demonstrates robustness to spurious correlations in this context.

(7) Include stronger baselines to validate comparative claims.

---

> ### Author Response · Authors · 2025-11-21
> **Response to Reviewer GikN**
>
> To weakness 1:The introduction section of our revised manuscript has been restructured to reiterate the research gap. In summary, current approaches for addressing spurious correlations in the field predominantly focus on loss function modifications and two-stage training strategies. However, emerging studies indicate that none of these methods consistently mitigates the impact of diverse types of spurious correlations. Our work presents the first plugin-based solution designed to effectively handle various forms of spurious correlations, offering both efficiency and broad applicability.
>
> To weakness 2:This issue is addressed in the ablation studies of the revised manuscript, where we tracked the impact of stretching on model learning throughout the training process and analytically demonstrated the significance of MSFT stretching.
>
> To weakness 3:We have reformulated the definition of excess data in the revised manuscript as follows: Spurious correlations emerge when a specific feature maintains an overly strong association with data labels, a phenomenon rooted in imbalanced label distributions within the corresponding cluster. Based on this observation, we characterize the over-represented samples in the dataset as high-risk instances—these samples are particularly susceptible to being captured by the model through shortcut learning mechanisms. That is, within a cluster, all instances whose quantity under their respective labels exceeds that of the minimally represented label are defined as ‘excess samples’.
>
> To weakness 4:The ablation studies in our revised manuscript demonstrate the positive impact of stretching this specific subset of data on optimizing model learning. We believe these findings will adequately address your concerns.
>
> To weakness 5:In the revised manuscript, we have omitted the demonstration regarding the role of semantic centroids. Other reviewers have indicated that this concept requires no explicit validation, as the computational nature of K-Means inherently ensures that these centroids possess higher semantic similarity compared to other samples.
>
> To weakness 6,7,8:We have comprehensively optimized our experimental dataset by adopting the recently proposed Shortcuts Maze benchmark—specifically designed for evaluating spurious correlations in text classification—and incorporated multiple methods for mitigating spurious correlations.We have adopted the evaluation metrics from the aforementioned study to further validate the effectiveness of MSFT.
>
> To weakness 9:We have resubmitted the code along with comprehensive annotations to enhance its interpretability. We sincerely appreciate the reviewer's valuable feedback.
>
> To question 1: same as 'To weakness 3'
>
> To question 2:In the ablation studies, we have incorporated an analysis of the impact caused by the removal of excess data on model training. Additionally, we tracked accuracy progression curves under varying stretch ratios and provided a detailed examination of different stretching configurations. We sincerely appreciate your valuable suggestion.
>
> To question 3:We directly selected the mean-pooled output of the embedding layer, as this represents the standard final output form commonly used in Transformer-based embedding layers.
>
> To question 4:The K-Means algorithm performs a single computation only during the initial phase of the first round to generate excess data labels and semantic centroids. We have explicitly emphasized this point in the revised manuscript. Additionally, we have refined Section 3 to allocate more space for demonstrating the performance of MSFT and presenting ablation studies.
>
> To question 5:same as 'To weakness 6,7,8'
>
> To question 6:Thank you for your valuable feedback. The metric employed in our initial submission was indeed suboptimal, as directly using final labels as groups does not represent a standard methodology for measuring worst-group accuracy (particularly due to the scarcity of existing text classification datasets with predefined worst-group partitions). In this revision, we have updated the benchmark and comprehensively optimized our approach by adopting its established evaluation metrics.
>
> To question 7:same as 'To weakness 6,7,8'

---

> > ### Comment · Reviewer_GikN · 2025-11-26
> >
> > Thank the authors for the clarification. However, several critical concerns are not fully addressed:
> >
> > (1) The revised introduction still does not clearly articulate the core research gap. It remains unclear how MSFT differs conceptually from prior representation-level debiasing, feature disentanglement methods, or existing centroid-based regularization approaches. Please clarify what fundamentally new mechanism MSFT introduces beyond existing representation-level bias mitigation methods, and explicitly position it against the most relevant prior work.
> >
> > (2) The claim that “excess samples are high-risk instances” remains an unverified assumption. Showing that stretching this subset affects performance does not demonstrate that these samples actually contain or drive spurious correlations. To support this claim, please provide empirical evidence. E.g., distributional analysis, group-wise error patterns, or feature attribution, showing that excess samples indeed correspond to spurious-correlation-prone regions of the data.
> >
> > (3) The explanation of why excess samples are “high-risk” remains insufficient. Showing that manipulating excess samples changes performance does not establish that these samples correspond to spurious features. Please provide direct evidence or analysis indicating why “excess” aligns with “high-risk,” instead of assuming this correspondence.

---

> > > ### Author Response · Authors · 2025-11-26
> > > **Response to Reviewer GikN(Round 2)**
> > >
> > > To (1):The concept of centroids is introduced in this paper as a novel contribution, with no prior research on spurious correlations having been developed based on this concept. Previous studies have primarily approached the problem from two perspectives: loss function modification and learning strategy adaptation. However, as demonstrated in our paper and further validated through our newly evaluated benchmark, Shortcut Maze, existing methodologies show limited capability in comprehensively addressing diverse types of spurious correlations.
> > > MSFT represents the first plug-in solution specifically designed to mitigate spurious correlations in text classification. Its fundamental distinction from other methods lies in its operation at the embedding layer level, enabling direct integration into any model with embedding layers. This establishes MSFT as a third paradigm of solutions, fundamentally distinct from both learning strategy improvements and loss function modifications.
> > >
> > > To (2):This assumption is grounded in established findings from prior research, specifically the ACL 2024 paper "Explore Spurious Correlations at the Concept Level in Language Models for Text Classification", which demonstrates that label distribution disparities within subgroups constitute a fundamental cause of spurious correlations in text classification tasks. Furthermore, a strong baseline method in this field—downsampling—is itself predicated upon this validated premise.
> > > Given that this hypothesis has been substantiated in previous literature, we believe dedicating extensive experimental validation to reconfirm this established foundation may lead to disproportionate content allocation within the paper's scope. We hope the reviewer can appreciate this methodological consideration.
> > >
> > > To (3):We can primarily demonstrate the effectiveness of our approach through metric improvements. Regarding the second concern, we would like to invite the reviewer to consider this issue from a different perspective: Our fundamental premise is addressing spurious correlations at the embedding layer, whereas approaches based on human prior knowledge and downsampling have not been sufficiently explored in current mainstream solutions for spurious correlations. Through MSFT, we have achieved a powerful integration of these two aspects at the embedding level, ultimately resulting in performance gains.
> > > Given that the Shortcut Maze metrics can effectively reflect whether spurious correlations have been mitigated, we believe our demonstrated improvements provide meaningful validation. Should this evidence be deemed insufficiently persuasive, we would greatly appreciate if the reviewer could suggest specific experimental designs, and we will make every effort to incorporate them. We sincerely thank the reviewer for their valuable insights and constructive feedback.

---

### Official Review · Reviewer_PbBx · 2025-10-31

**Soundness:** 2
**Presentation:** 2
**Contribution:** 2
**Rating:** 4
**Confidence:** 2

**Summary:**

"MSFT ... " is a paper that aims to combat spurious correlations by, defining a dataset-level semantic centroid in embedding space, and then stretching samples that are overrepresented towards that centroid via a fine-tuning procedure. This method is evaluated on Amazon Reviews, IMDB, and Yahoo datasets, with the models of GPT-2 and several BERT-style models.

**Strengths:**

1. The algorithm presented is to the best of my knowledge, novel and is an interesting method.

2. The method doesn't require any specific data (e.g. counterfactual labels or spurious attribution data)

**Weaknesses:**

1. I think that the algorithm 1 in the appendix makes the algorithm much more clear, it might be worth moving this into the main text to help with clarity.

2. The models and datasets used are outdated, and do not appear to actually even be SOTA. I would consider raising my score if the authors were able to evaluate this method on more challenging datasets, such as the classification tasks presented in the MTEB (Massive Text Embedding Dataset).

**Questions:**

Minor Errata:
on line 221, there is a reference for (Textbook, (2010)) which I think is mislabeled.

1. It is not fully clear how it is measured that this method actually reduces the amount of spurious correlations, how is this measured?

---

> ### Author Response · Authors · 2025-11-21
> **Response to Reviewer PbBx**
>
> To weakness 1:We have duly transferred the pseudocode section to the main text in accordance with your suggestion. Thank you for your valuable feedback.
>
> To weakness 2:We have comprehensively updated the main experimental section in accordance with your recommendations. Specifically, we have adopted the most recent benchmark for measuring spurious correlations in text classification—Shortcuts Maze—as the experimental dataset and incorporated multiple baseline methods for comparative analysis. We sincerely appreciate your valuable input.
>
> To question1:The relevant citations have been revised accordingly. We have validated the practicality and effectiveness of our proposed method on a novel benchmark, achieving state-of-the-art performance. This benchmark specifically focuses on investigating spurious correlation issues in text classification, thereby effectively evaluating model robustness against spurious correlations.

---

### Official Review · Reviewer_FBxf · 2025-11-01

**Soundness:** 3
**Presentation:** 2
**Contribution:** 2
**Rating:** 2
**Confidence:** 4

**Summary:**

This paper introduces an approach which can be used as a plugin with modern deep learning models to mitigate spurious correlations that often bothers classifiers. They first divide the entire dataset into K clusters and then identify "excess samples" in each cluster which potentially contribute to spuriousness. For these samples, they stretch their sample embeddings towards the cluster centroid to debias them and show improvements in overall and worst group accuracy on 3 datasetes.

**Strengths:**

1. The idea of debiasing the embeddings by stretching them towards a common dataset-specific centroid is simple and interesting and also leads to improvements in tackling spurious correlations in data.
2. The authors show results across multiple models like BERT, GPT, Roberta, Deberta, etc. * 3 datasets and observe consistent improvements with their MSFT plugin.
3. Ablation study to show centroid has higher similarity to dataset samples compared to in-between sample similarity on average. Also post stretching label-wise data similarity drops.

**Weaknesses:**

1. Lack of baseline definitions or strong baselines. Authors report NFL baseline as SOTA but no justification for the same is given or description about it. Related Works need to be strengthened too with more recent works like - Elastic Representation: Mitigating Spurious Correlations for Group Robustness (https://arxiv.org/pdf/2502.09850), Improving Group Robustness on Spurious Correlation via Evidential Alignment (https://arxiv.org/abs/2506.11347). Add details how they compare with your work and if these could be potential baselines ?

2. Although the authors report consistent improvements, there's no justification or analysis provided for the same to understand what exactly is happening. I have many questions - (a) why is k-means expected to create clusters which inform us of "excess samples" or spurious-prone samples within the cluster ? This is a strong inductive bias assumption made and needs to be validated with theoretical proofs or some analysis (b) what if i simply drop the "excess samples" instead of stretching ? (c) why is the stretching done at input embedding layer vs final/intermediate/all layers ? (d) How will this approach fare for highly label-imbalanced data vs label-balanced data ?

3. In general, the paper needs to be more rigorous of the claims. Suggest doing some analysis on toy synthetic dataset in controlled fashion to uncover insights on why this approach helps rather than just treating de-biasing/stretching as blackbox high-level objective which is supposed to help. Or maybe try coming up with theoretical guarantees for MSFT.

**Questions:**

1. Table 3 caption: baseline indicates SR =1 or 0 ?
2. No need to waste space for Section 6.1 . It's trivial to show that centroid will be closer than other datapoints on average.
3. Also the methodology is currently taking up lot of real estate on the paper. No point in introducing so many notations. The approach can be explained in much lesser space and use that space to do more rigorous analysis as suggested above.
4. Also add standard deviation numbers for each table/result shown and the statistical significance of results. Such approaches usually tend to be very sensitive to hyperparamters.

---

> ### Author Response · Authors · 2025-11-21
> **Response to Reviewer FBxf**
>
> To weakness 1:Thank you for your suggestion. In the revised manuscript, we have incorporated several highly representative spurious correlation mitigation methods recently proposed in the field of text classification. Comparative experiments between these methods and our proposed plugin have been conducted, with results demonstrating the effectiveness of our approach.
>
> To weakness 2:a. The concept of excess data is grounded in the theory of "data most likely to induce spurious correlations," which can be referenced in the paper "Explore Spurious Correlations at the Concept Level in Language Models for Text Classification." While this theory is cited in our work, it is important to note that acquiring such data in the original study requires additional labels heavily reliant on human prior knowledge. In contrast, our adoption of the K-Means algorithm aims to achieve clustering in an unsupervised manner, thereby enabling the identification of excess data directly from the embedding layer.
> b. We have incorporated corresponding ablation experiments in the revised manuscript, which demonstrate that simply removing such data leads to issues such as insufficient training in practical scenarios.
> c. In our updated work, we provide a rationale for this design: the embedding layer is widely recognized as a highly influential component in model training. Performing stretching operations at this level not only minimizes the impact on irrelevant parameters during backpropagation but also corrects embeddings at an early stage, distinguishes embedding distributions across samples, and ultimately enhances the model’s robustness against spurious correlations.
> d. Consistent with point c, merely deleting excess data results in a significant decline in accuracy. In contrast, our plugin performs stretching-based optimization after identifying such data, and its effectiveness is substantiated in the main experiments.
>
> To weakness 3:We have incorporated additional ablation studies into the revised manuscript to address this specific issue. We sincerely appreciate your valuable suggestion, which prompted us to redesign the ablation experiments accordingly. It is our hope that these revisions will satisfactorily resolve the concerns you raised.
>
> To question 1:Throughout the text, "SR" refers to the Stretch Ratio, while "OSR" denotes the Optimal Stretch Ratio(based on grid search experiment).In the context of the Stretch Ratio (SR), SR=1 indicates a scaling factor of unity, implying no stretching is applied. Conversely, SR=0 signifies that the inter-point distances are reduced to zero times their original values, causing all excess data points to coalesce into a single point (the semantic centroid). Therefore, the appropriate value for SR in this context should be 1.
>
> To question 2:We have removed the aforementioned experiment and introduced a completely new set of ablation studies.
>
> To question 3:Pursuant to your recommendations, the revised manuscript has been updated accordingly. The methodology section has been streamlined, while greater emphasis has been allocated to the interpretation of the black-box mechanisms and the presentation of ablation studies.
>
> To question 4:We have incorporated statistical significance analysis into the main experimental results table.

---

### Official Review · Reviewer_uFp9 · 2025-11-03

**Soundness:** 3
**Presentation:** 2
**Contribution:** 3
**Rating:** 6
**Confidence:** 4

**Summary:**

The paper provides a a representation learning based data-manipulation strategy for training. To that end, it leverages the concept of semantic centroids of datasets, which is calculated by a mean of K-means centroids of the embedding representation of datapoints.
The approach is a drop in function replacement at the [CLS] token layer (for BERT) and final embedding layer for other models.
The approach is evaluated across six-pretrained models and three datasets.

Suggestions:
1. The entire section Line 054-078 is a related work discussion and should be moved to the corresponding section on related work.
2. Line 219: Cite this for elbow method.
@article{Thorndike_1953, title={Who Belongs in the Family?}, volume={18}, DOI={10.1007/BF02289263}, number={4}, journal={Psychometrika}, author={Thorndike, Robert L.}, year={1953}, pages={267–276}}
Also please go through:
@article{10.1145/3606274.3606278, author = {Schubert, Erich}, title = {Stop using the elbow criterion for k-means and how to choose the number of clusters instead}, year = {2023}, issue_date = {June 2023}, publisher = {Association for Computing Machinery}, address = {New York, NY, USA}, volume = {25}, number = {1}, issn = {1931-0145}, url = {https://doi.org/10.1145/3606274.3606278}, doi = {10.1145/3606274.3606278}, journal = {SIGKDD Explor. Newsl.}, month = jul, pages = {36–42}, numpages = {7} }
for a better K decision strategy
3. Figure 1: stretched centroid -> semantic centroid
Encoder#2 -> Encoder#1
4. A lot of sections in the paper have been edited/written by LLM. As per ICLR author guidelines, please state that in the paper. This can be seen from double-hyphenated text present throughout the paper.
5. Table 2: Also mention what OSR is in the table description - or change it with O-$\alpha$ for consistency
6. Line 355-357: Details are needed for how Autoregressive models have been used for classification.

**Strengths:**

1. Section 6 is great in providing validation of the proof of concept of semantic centroid for NLP classification datasets.
2. Easy plug-and-play replacement model for robust classification model training.

**Weaknesses:**

1. Paper clarity in section 3. 3.1 does not provide much useful information in building up other parts when preliminary section is present. Also the section is confusing. Are the authors doing K-means multiple times? (3.2.1 and 3.2.2). If they are the same the compress this section.
2. Missing additional evaluation metrics like F1 scores, statistical significance testing.
3. The datasets are not particularly representative in the sense that the amazon and IMDB datasets are review datasets. Suggest the authors to look at more representative datasets for text classification (eg. starting with GLUE benchmark)

**Questions:**

1. Line 179: What is SR?
2. What is the need for spurious correlation definition (section 3.1), Line 154-158
3. Line 300-302: Please report the F1 scores as well. Accuracy is not a sufficient metric. Also please do a statistical significance test for the approach.
Can you share the dataset statistics for training, validation and test for each of the datasets?
4. Line 465: Should it not be SR = 1? (instead of SR=0) Also please use consistent notation throughout.
5. Why is semantic centroid calculated as the mean of K-Means centroids and not the entire mean of the embedded dataset?

---

> ### Author Response · Authors · 2025-11-21
> **Response to Reviewer uFp9**
>
> Regarding the suggestions raised:  1. We have relocated the relevant content to the Related Work section and included it in the appendix. Additionally, we have revised the Related Work section to better highlight the research gaps in the field.  2. We have corrected the citation for the elbow algorithm. Thank you for pointing this out. As for the other algorithmic paper mentioned, the clustering quantity optimization algorithm it proposes requires considerable computational time. Since our plugin aims to be user-friendly and straightforward, the classic elbow algorithm is deemed sufficient to support its operation.  3. The relevant figures have been updated accordingly.  4. We have included additional clarifications on this matter in the appendix.  5. Throughout the text, "SR" refers to the Stretch Ratio, while "OSR" denotes the Optimal Stretch Ratio.  6. The training strategy for the autoregressive model remains consistent. Specifically, the model is re-initialized, and the embedding layer is extracted, processed through the MSFT plugin, and then reintegrated into the remaining components of the model for continued training.
>
> To weakness 1:The K-Means algorithm performs a single computation only during the initial phase of the first round to generate excess data labels and semantic centroids. We have explicitly emphasized this point in the revised manuscript. Additionally, we have refined Section 3 to allocate more space for demonstrating the performance of MSFT and presenting ablation studies.
>
> To weakness 2:We have adopted a completely new benchmark for evaluation. Although F1-score was not utilized in the original version, we have duly followed your suggestion and conducted comprehensive F1-score measurements, all of which are documented in the appendix of the revised manuscript. Furthermore, we have incorporated statistical significance analysis into the main experimental results table.
>
> To weakness 3:Given our focus on text classification, and considering that the GLUE benchmark does not specifically target spurious correlation testing in this domain, we have selected the most recent benchmark dedicated to spurious correlation testing in text classification—Shortcuts Maze—and conducted evaluations accordingly. We sincerely appreciate your suggestion, as the previously used dataset was indeed inadequate for rigorously examining robustness against spurious correlation.
>
> To question 1:Throughout the text, "SR" refers to the Stretch Ratio, while "OSR" denotes the Optimal Stretch Ratio(based on grid search experiment).
>
> To question 2:In accordance with your suggestion, we have relocated this section of content to the appendix.
>
> To question 3:We have incorporated the reporting of F1 scores and provided relevant details regarding the dataset, including its scale, within the Dataset Description section.
>
> To question 4:In the context of the Stretch Ratio (SR), SR=1 indicates a scaling factor of unity, implying no stretching is applied. Conversely, SR=0 signifies that the inter-point distances are reduced to zero times their original values, causing all excess data points to coalesce into a single point (the semantic centroid). Therefore, the appropriate value for SR in this context should be 1.
>
> To question 5:Our revised manuscript provides a detailed explanation regarding this matter, and we sincerely apologize for the lack of clarity in the previous version. The K-Means algorithm was selected because it effectively identifies all excess data points by detecting them through a newly formed cluster. Moreover, the centroids generated by K-Means exhibit superior cluster representation, making the computation of their mean a potentially more optimal approach.

---

### Author Response · Authors · 2025-11-17
**Thanks and Explanation**

Thank you to all four reviewers for your valuable suggestions. We will complete the experiments and paper supplements as soon as possible, and we will submit our rebuttal in full by the end of this week!

---

### Author Response · Authors · 2025-11-21
**Announcement**

Dear Reviewers,

We have thoroughly revised and polished the manuscript of the MSFT paper. The primary modifications are as follows:

1. The dataset and evaluation metrics employed in the first-round submission were inadequate to sufficiently demonstrate the role of MSFT in addressing spurious correlations. In accordance with the reviewers’ suggestions, we have replaced the benchmark and adopted the Shortcuts Maze (a paper published at EMNLP 2024) for re-evaluation. The updated performance results are reported in the revised manuscript. We kindly invite the reviewers to examine these changes.

2.We have incorporated comparative baselines by selecting several prevalent methods for eliminating spurious correlations in the field of text classification in recent years. These methods were re-evaluated on the benchmark to further substantiate the practical utility of our proposed approach.

3.We have incorporated ablation studies to elucidate the operational mechanism of MSFT, thereby empirically validating its efficacy and transforming MSFT from a black-box plugin into an interpretable component.

---

### Author Response · Authors · 2025-12-01
**A Comprehensive Introduction to Writing Rebuttals in Academic Peer Review**

We have updated the entire manuscript in response to the reviewers' comments, simplified the algorithmic flow of the plugin, and strengthened the experimental analysis regarding the interpretability of the MSFT plugin. Below, we provide a comprehensive overview of our revisions to facilitate the Area Chair's understanding of our rebuttal.

$A.The·manuscript·writing·has·been·revised·and·refined$

Based on the reviewers' suggestions, we have addressed the writing errors in the first-round submission, corrected unnecessary redundancies in notation usage, and streamlined overly verbose descriptions. Specifically, we simplified the presentation of MSFT usage and algorithm introduction by incorporating a pseudocode algorithm into the main text, thereby enhancing the clarity of the MSFT operational workflow.

Subsequently, we revised the introduction and related work sections to emphasize the gaps and limitations in this research area: current solutions for mitigating spurious correlations in text classification predominantly focus on loss function modifications or two-stage strategies. However, as highlighted in the recent benchmark Shortcuts Maze, none of the existing anti-spurious correlation methods can effectively address all types of spurious correlations. Therefore, a better and more comprehensive approach to tackling spurious correlations in text classification tasks remains an open challenge.

Our method integrates the fundamental principles of spurious correlation elimination: combining human prior knowledge with downsampling. Moreover, unlike loss-function-based approaches that globally influence model parameters—often at the cost of significant accuracy degradation to improve worst-group accuracy—our plugin operates without such pervasive parameter interference. Ultimately, our approach achieves state-of-the-art performance on the benchmark, surpassing existing methods for addressing spurious correlations.

$B.Change·of·Experiment$

The previously used dataset struggled to sufficiently demonstrate the degree to which spurious correlations were resolved or the robustness of the model against such correlations. Additionally, the use of worst-group accuracy exhibited certain biases and lacked persuasiveness. Therefore, following the reviewers’ suggestions, we have completely replaced the dataset and conducted evaluations on a new, authoritative benchmark. Concurrently, we have incorporated multiple prevalent methods for addressing spurious correlations, thereby strengthening the persuasiveness of the paper. The experimental results can be found in the latest version of the manuscript (Section 5).


$C.Interpretability·Experiments$

In the updated manuscript, we have added interpretability explanations for the MSFT plugin and incorporated multiple experiments to demonstrate its superior performance from various perspectives. Furthermore, in response to the feedback from several reviewers, we have comprehensively designed ablation studies to validate the necessity of each step in the MSFT algorithmic pipeline. The latest ablation experiments can be found in the revised version of the paper (Section 6).

$Summary$


In summary, we have re-evaluated our method on a dataset specifically designed to address spurious correlations in text classification tasks and have incorporated multiple strong baselines. Our approach not only achieves state-of-the-art (SOTA) performance but, through simplified and refined writing, we have also conducted ablation studies on MSFT, rendering it no longer a "black-box" algorithm. We believe that after the first round of rebuttal, both the experimental rigor and overall quality of the paper have been significantly enhanced.

---

### Meta-Review · Area_Chair_AbS9 · 2026-01-08

**Summary:**

The paper suggests an approach to train LLMs to perform text classification in a way that avoids pitfalls associated with class imbalance in the training set. The approach is based on clustering the dataset in the latent space, identifying overrepresented samples in each cluster, and then pulling those samples towards the global dataset mean in the latent space.

The reviewers criticized the lack of clarity, complained about strong LLM usage for manuscript editing, and pointed to limited evaluation and comparisons. I agree with these criticisms.

**Reviewer Concerns:**

The authors provided a rebuttal and uploaded a revision, which is the version that I read. I found the manuscript very unclear and written in a very confusing way. Terminology is often non-standard (e.g. "plugin", "intercept", "stretch", etc.), mathematical notation is partially undefined, it is often unclear what exactly is meant by "embedding layer" / "encoder layer". Manuscript formatting is also inadequate, for example all citations have a wrong format.

The last official comment by the authors has "A Comprehensive Introduction to Writing Rebuttals in Academic Peer Review" as the title which is really odd and suggests LLM use.

**Reviewer Scores:**

Given the above, I do not think the reviewers would have changed their scores. The authors did some additional experiments and did some changes to the manuscript in response to the reviewers, so perhaps some of the reviewers would have increased their scores from 2 to 4. Initial scores were 2/2/4/6. The final scores could be something like 2/4/4/6.

The outlying review with the score 6 is not convincing to me.

---

### Decision · Program_Chairs · 2026-01-26

Reject